

# TempestExtremes v1.0: A Framework for Scale-Insensitive Pointwise Feature Tracking on Unstructured Grids

Paul A. Ullrich[1] and Colin M. Zarzycki[2]

[1] Department of Land, Air and Water Resources, University of California Davis, Davis, CA, USA.

[2] National Center for Atmospheric Research, Boulder, CO, USA.

*Correspondence to:* Paul A. Ullrich
(paullrich@ucdavis.edu)

**Abstract.** This paper describes a new open-source software framework for automated pointwise feature tracking that is applicable to a wide array of climate datasets using either structured or unstructured grids. Common climatological pointwise features include tropical cyclones, extratropical cyclones and tropical easterly waves. To enable support for a wide array of detection schemes, a suite of algorithmic kernels have been developed that capture the core functionality of algorithmic tracking routines

from throughout literature. A review of efforts related to pointwise feature tracking from the past three decades is included. Selected results using both reanalysis datasets and unstructured grid simulations are provided.

## 1 Introduction

Automated pointwise feature tracking is an algorithmic technique for objective identification and tracking of meteorological features such as extratropical cyclones, tropical cyclones and tropical easterly waves, and has emerged as an important and

desirable data processing capability in climate science. Software tools for feature tracking – typically referred to as "trackers" – have been employed to evaluate model performance and answer pressing scientific questions regarding anticipated changes in atmospheric features under climate change. Exploration of tracker literature has exposed a breadth of potential techniques that have been applied to climate datasets with varied spatial resolution and temporal frequency (a comprehensive review of the tracking literature can be found in appendices A, B and C). Nonetheless, the definition of an optimal objective criteria for

key atmospheric features has eluded development, and ambiguity in the formal definition of these features suggests that there may be no singular criteria capable of both perfect detection and zero false alarm rate. Further, as observed by Walsh et al. (2007) and Horn et al. (2014) for tropical cyclones and Neu et al. (2013) for extratropical cyclones, feature tracking schemes can produce wildly varying results depending on the specific choice of threshold variables and values. Therefore, we argue that uncertainties associated with objective tracking criteria should be addressed with an ensemble of detection thresholds

and variables, whereas blind application of singular tracking formulations should be avoided. To this end, it is the goal of this paper to review the vast literature on trackers and use this information to inform the development of a unified framework (TempestExtremes) that enables a variety of tracking procedures to be quickly and easily applied across across arbitrary spatial





resolution and temporal frequency. This manuscript focuses largely on the technical aspects of pointwise feature tracking, but sets the stage for future studies on parameter sensitivity and optimization.

Most algorithmic Lagrangian trackers of pointwise features (such as cyclones and eddies) share a common procedure:

1. Identify an initial set of candidate points by searching for local extrema. Local extrema can be further specified, for instance, by requiring that the they be sufficiently anomalous when contrasted with their neighbors. For most cyclonic structures, either minima in the sea level pressure field or maxima in the absolute value of the relative vorticity are used.

2. Eliminate candidate points that do not satisfy a prescribed set of thresholds. For instance, tropical cyclones typically require the presence of an upper-level warm core that is sufficiently near the sea level pressure minima that defines the storm center. Additional criteria, such as a minimum threshold on relative vorticity can be used to eliminate spurious detections.

3. Connect candidate points together in time (referred to as *stitching*) to generate feature paths, eliminating paths that are of insufficient length or do not meet additional criteria.

Although the actual implementations of this procedure does vary throughout the literature, a review of this material reveals several core algorithms (kernels) that are common across implementations. Based on our analysis, the five most commonly employed kernels are as follows:

– Computation of anomalies in a data field from a spatially averaged mean.

– Identification of local extrema in a given 2D data field (for instance, sea level pressure minima).

– Determination of whether a closed contour exists in a data field around a particular point.

– Determination of whether, in the neighborhood of a particular point, a data field satisfies a given threshold.

– Stitching of candidates from sequential time slices to build feature tracks.

The development of a robust implementation of these five kernels will be the focus of the remainder of this paper.

Feature tracking that is robust across essentially arbitrary datasets requires some additional considerations. Detection criteria and thresholds for tracking are often tuned based on the characteristics of a particular dataset, such as temporal resolution, spatial resolution and regional coverage. Unfortunately, this has led to an abundance of schemes that often cannot be directly compared, or applied in a more general context. To this end, we focus on kernels that are insensitive to the characteristics of the input data. For instance, averaging or searching over a discrete number of grid points around each candidate (a common approach) is incompatible with scale insensitivity since the physical search radius would be dependent on the spatial resolution of the data. Identification of local extrema is also a resolution-sensitive procedure, since the number of extrema will often scale with the number of spatial data points – however, a closed contour criteria based on a physical distance is largely resolution-insensitive. To achieve robust applicability, a general framework should:





...use great-circle arcs for all distance calculations. This avoids issues associated with latitude-longitude distance that emerges near the poles.

...support structured and unstructured grids. This eliminates the need for post-processing of large native-grid output files and enables detection and characterization simultaneous with the model execution.

...not contain hard-coded variable names, so as to ensure robust applicability across reanalysis datasets and applicability to a variety of problems.

...allow for easy intercomparison of detection schemes by enabling detection criteria and thresholds that are compactly specified on the command line.

Well-known automated software trackers include TRACK (Hodges, 1994, 1995, 2015) and the Geophysical Fluid Dynamics

Laboratory (GFDL) TSTORMS package (Vitart et al., 1997; Zhao et al., 2009). Both of these software packages have been used extensively to examine pointwise features in the atmosphere, but do not completely satisfy the four requirements above.

The remainder of the paper is organized as follows: Section 2 describes the algorithms and kernels that have been implemented in the TempestExtremes software framework. Selected examples of tropical cyclone, extratropical cyclone and tropical easterly wave detection are then provided in section 3, followed by conclusions in section 4. The appendices provide a review of

relevant literature on pointwise feature trackers of extratropical cyclones (appendix A), tropical cyclones (appendix B) and tropical easterly waves (appendix C). A technical guide to the use of the TempestExtremes tools `DetectCyclonesUnstructured` and `StitchNodes` is provided in appendices D and E.

## 2   TempestExtremes Algorithms and Kernels

This section describes the key building blocks that have been developed in constructing our detection and characterization

framework. Pseudocode is utilized throughout to describe the structure of each algorithm.

### 2.1   Unstructured grid specification

For purposes of determining connectivity of the unstructured grid, we require the specification of a node graph (one such node graph is depicted in Figure 1). The connectivity information is stored textually as an adjacency list via a variable-length comma-separated variable file. The total number of nodes ($N$) is specified at the top of the file, followed by $N$ lines containing

the longitude (lon), latitude (lat), associated area, number of adjacent nodes, and finally a 1-indexed list of all adjacent nodes, such as depicted below:





```
<total number of nodes>
<lon>,<lat>,<area>,<# adj. nodes>,<first adj. node>,..,<last adj. node>
...
```

## 2.2 Great-circle distance

As mentioned earlier, in order to avoid sensitivity of the detection scheme to grid resolution, great-circle-distance has been employed throughout. In terms of regular latitude-longitude coordinates, the great-circle-distance ($r$), for a sphere of radius $a$,

between points $(\lambda_1, \varphi_1)$ and $(\lambda_2, \varphi_2)$ is defined via the symmetric operation

$$r(\lambda_1, \varphi_1; \lambda_2, \varphi_2) = a \arccos\left(\sin\varphi_1 \sin\varphi_2 + \cos\varphi_1 \cos\varphi_2 \cos(\lambda_1 - \lambda_2)\right). \tag{1}$$

Algorithmically, this calculation is implemented as `gcdist(p,q)` for given graph nodes `p` and `q`.

## 2.3 Efficient neighbor search using $k$-d Trees

Three-dimensional ($k = 3$) $k$-d trees (Bentley, 1975) are used throughout our detection code using the implementation of

Tsiombikas (2015). Although $k$-d trees use 3D straight-line instance instead of great-circle-distance, we utilize the observation that straight-line and great-circle distance maintain the same ordering for points confined to the surface of the sphere. Three key functions made available by the $k$-d tree structure are used:

`K = build_kd_tree(P)` constructs a $k$-d tree `K` from a point set `P`.

`q = kd_tree_nearest_neighbor(K, p)` locates the nearest neighbor `q` to point `p` using the $k$-d tree `K`.

`S = kd_tree_all_neighbors(K, p, dist)` locates all points that are within a distance `dist` of a point `p` within the $k$-d tree `K`.

A key advantage of $k$-d trees is their relatively efficient $O(n \log n)$ construction time and $O(\log n)$ average time nearest neighbor search, and $O(n)$ data storage requirements.

## 2.4 Computing a spatial averaged mean

Many existing tracking algorithms use either a spatially-averaged mean field or an anomaly field computed against the spatially-averaged mean (Haarsma et al., 1993; Bengtsson et al., 1995). The mean operation (implemented in TempestExtremes as `_MEAN()` in the variable specification) is computed on unstructured grids via graph search (see Algorithm 1). Anomaly from the mean can then be computed in conjunction with the `_DIFF` operator (see Appendix D2).



## 2.5 Extrema detection

For purposes of computational efficiency, candidate points are initially located by identifying local extrema in a given field (for instance SLP) via `find_all_minima` (Algorithm 2). Candidates are then eliminated if they are "too close" to stronger extrema (Algorithm 3) (e.g. Pinto et al. (2005)). The initial search field is specified to TempestExtremes either via the `--searchbymin` or `--searchbymax` command line argument. The merge distance used in `merge_candidates_minima` is specified via the `--mergedist` command line argument.

## 2.6 Closed contour criteria

Although a first pass at candidate points may be made by looking for local extrema (comparing against all neighboring nodes), this criteria is not robust across model resolution. That is, the distance between a node and its neighbors decreases proportional to the local grid spacing, and so does not define a "physical" criterion. Consequently, we instead advocate for a *closed contour criteria* to define candidate nodes. Closed contours were first employed by Bell and Bosart (1989), who used a 30m 500 hPa geopotential height contour to identify closed circulation centers. Their approach used radial arms generated at 15° intervals over a great-circle-distance of 2° and required that geopotential heights rise by at least 30m along each arm. Unfortunately, the use of radial arms to define the closed contour is again sensitive to model resolution, since it has the potential to only sample as many neighbors as radial arms employed.

Here we propose an alternative closed contour criteria that is largely insensitive to model resolution using graph search to ensure that all paths along the unstructured grid from an initial location `p0` lead to a sufficiently large decrease (or increase) in a given field `G` before reaching a specified radius. This criteria is illustrated in Figure 2, and is implemented in Algorithms 4 and 5 (for closed contours around local maxima). The closed contour criteria is implemented in TempestExtremes via the command line argument `--closedcontourcmd`. An analogous command line argument `--noclosedcontourcmd` is also provided, which has similar functionality but discards candidates that satisfy the closed contour criteria (this may be desirable, for instance, to identify cyclonic structures that do not have a warm core).

## 2.7 Thresholding

Additional threshold criteria may be applied at the detection stage in order to further eliminate undesirable candidates. For example, a common threshold criteria requires that a field `G` satisfy some minimum value within a distance `dist` of the candidate, as implemented in Algorithm 6. TempestExtremes implements thresholding via the command line argument `--thresholdcmd`.

## 2.8 Stitching

The basic track stitching procedure (which represents the Reduce() stage in MapReduce) is implemented in Algorithm 7 using the output from the detection procedure at each time level (stored in set array `P[1..T]`). It requires additional parameters to specify a maximum great-circle-distance between nodes (`dist`), and a maximum gap size (`maxgap`). Here, gap size refers





to the maximum number of sequential non-detections that can occur before a path is considered terminated. This argument is useful, for instance, for tracking tropical storms that temporarily weaken below acceptable criteria before re-strengthening.

For simplicity, $k$-d trees are constructed at each time level in order to maximize the efficiency of the search. Each candidate pair (time, node) can only be used in one path, and so construction simply requires exhausting the list of available candidates. Once paths have been constructed, additional criteria can be applied – for instance, minimum path length or additional criteria based on minimum path length or minimum distance between the start and endpoints of the path (see Appendix E). Thresholds based on field values may also be applied, *e.g.* wind speed must be greater than a particular value for at least 8 time steps of each track.

### 2.9 Parallelization Considerations

Feature tracking fits well into a general framework known as MapReduce (Dean and Ghemawat, 2008), which is a combination of a Map(), an embarrassingly parallel candidate identification procedure applied to individual time slices, and a Reduce(), which stitches candidates across time to build feature tracks. A key advantage of employing this framework is that substantial work has been undertaken to understand optimal strategies for parallelization of MapReduce-type algorithms (e.g. Prabhat et al. (2012)) in order to mitigate bottlenecks associated with I/O and load balancing. TempestExtremes currently implements a simple parallelization strategy via MPI, although future work on this issue is forthcoming. As a timing example, TempestExtremes with MPI (16 tasks) finds and tracks tropical cyclones in 10 years of six-hourly climate data on a 0.5° latitude-longitude grid in an average of 3.8 minutes on the National Center for Atmospheric Research's (NCAR's) Yellowstone supercomputer.

## 3 Selected examples

Several selected examples are now provided. The first three examples use data from the NCEP Climate Forecast System Reanalysis (CFSR), available at 0.5° global resolution with 6-hourly output from 1979-present (Saha et al., 2010). The remaining example uses a custom variable-resolution simulation (Zarzycki and Jablonowski, 2014) (6-hourly output on a 110km base domain that is refined to 28km in the northern Atlantic and Pacific ocean basins) on both the native grid data and the regridded latitude-longitude grid data.

### 3.1 Tropical cyclones in CFSR

Our first example employs TempestExtremes for tropical cyclones (defined here as a cyclonic structure with a distinct warm-core). The command line we use to detect tropical cyclone-like features in CFSR is provided below. Climate data is drawn from three files denoted `$uvfile` (containing zonal and meridional velocities), `$tpfile` (containing temperature and pressure information) and `$hfile` (containing topographic height). Three-dimensional (time + 2D space) hyperslabs of CFSR data have been extracted, with `TMP_L100` corresponding to 400 hPa air temperature, and `U_GRD_L100` and `V_GRD_L100` corresponding to 850 hPa zonal and meridional wind velocities. Candidates are initially identified by minima in the sea level pressure (`PRMSL_L101`), and then eliminated if a more intense minimum exists within a great-circle-distance of 2.0°. The





closed contour criteria is then applied, requiring an increase in SLP of at least 200 Pa (2 hPa) within 4° of the candidate node, and a decrease in 400 hPa air temperature of 0.4 K within 8° of the node within 1.1° of the candidate with maximum air temperature. Since CFSR is on a structured latitude-longitude grid, the output format is `i,j,lon,lat,psl,maxu,zs`, where `i, j` are the longitude and latitude coordinates within the dataset, `lon, lat` are the actual longitude and latitude of the

5 candidate, `psl` is the SLP at the candidate point (equal to the maximum SLP within 0° of the candidate), `maxu` is the vector magnitude of the maximum 850 hPa wind within 4° of the candidate, and `zs` is the topographic height at the candidate point.

```
./DetectCyclonesUnstructured
  --in_data "$uvfile;$tpfile;$hfile" --out $outf
  --searchbymin PRMSL_L101 --mergedist 2.0
  --closedcontourcmd "PRMSL_L101,200.,4,0;
    TMP_L100,-0.4,8.0,1.1"
  --outputcmd "PRMSL_L101,max,0;
    _VECMAG(U_GRD_L100,V_GRD_L100),max,4;
    HGT_L1,max,0"
```

15 All outputs from DetectCyclonesUnstructured are then concatenated into a single file containing candidates at all times (`pgbhnl.dcu_tc_all.dat`). Candidates are then stitched in time to form paths, with a maximum distance between candidates of 8.0° (great-circle-distance), consisting of at least 8 candidates per path, and with a maximum gap size of 2 (most consecutive timesteps with no associated candidate). Because localized shallow low-pressure regions that are unrelated to tropical cyclones can form as a consequence of topographic forcing, we also require that for at least 8 time steps the underlying

20 topographic height (`zs`) be at most 100 meters. The associated command line for StitchNodes is:

```
./StitchNodes
  --in pgbhnl.dcu_tc_all.dat
  --out pgbhnl.dcu_tc_stitch.dat
  --format "i,j,lon,lat,psl,maxu,zs"
  --range 8.0 --minlength 8 --maxgap 2
  --threshold "zs,<=,100.0,8"
```

Once the complete set of tropical cyclone paths has been computed, total tropical cyclone counts over each 2° grid cell are plotted in Figure 3. The results show very good agreement with reference fields (Gray, 1968; Knapp et al., 2010).

## 3.2 Extratropical cyclones in CFSR

30 For our second example, we are interested in tracking extratropical cyclone features (defined by a cyclonic structure with no distinct warm-core). The command line we have used to detect cyclonic features without the characteristic warm-core of tropical cyclones (here referred to as extratropical cyclones) is given below. The command is identical to the TC detection configuration specified in section 3.1, except requiring that the feature does not possess a closed contour in the 400 hPa temperature field (no warm core).



```
./DetectCyclonesUnstructured
  --in_data "$uvfile;$tpfile;$hfile" --out $outf
  --searchbymin PRMSL_L101 --mergedist 2.0
  --closedcontourcmd "PRMSL_L101,200.,4,0"
  --noclosedcontourcmd "TMP_L100,-0.4,8.0,1.1"
  --outputcmd "PRMSL_L101,max,0;
     _VECMAG(U_GRD_L100,V_GRD_L100),max,4;
     HGT_L1,max,0"
```

Stitching is similarly analogous to section 3.1, except using a slightly more strict criteria on the underlying topographic height. The topographic filtering proved necessary in order to adequately filter out an abundance of topographically-driven low pressure systems, particularly in the Himalayas region. The command line used for stitching is given below:

```
./StitchNodes
  --in pgbhnl.dcu_tc_all.dat
  --out pgbhnl.dcu_tc_stitch.dat
  --format "i,j,lon,lat,psl,maxu,zs"
  --range 8.0 --minlength 8 --maxgap 2
  --threshold "zs,<=,70.0,8"
```

Once the complete set of extratropical cyclone paths has been computed, total extratropical cyclone density over each 2° grid cell is plotted in Figure 4. Although not extensively verified, the qualitative density of extratropical cyclones is well within the range of results from different trackers, as given by Neu et al. (2013).

### 3.3 Tropical easterly waves in CFSR

Tropical easterly waves are our third example of a pointwise feature that has been assessed in the tracking literature. In this example, northern hemisphere easterly waves (associated with positive relative vorticity) are tracked separately from southern hemisphere easterly waves (associated with negative relative vorticity) be instantiating `DetectCyclonesUnstructured` and `StitchNodes` twice and combining the resultant track files. All tracking is performed on the 600 hPa relative vorticity field, using relative vorticity maxima for northern hemisphere waves and relative vorticity minima for southern hemisphere waves. Since CFSR only provides absolute vorticity, relative vorticity must first be extracted by taking the difference between absolute vorticity and the planetary vorticity (the Coriolis parameter). This is done on the command line via `_DIFF(ABS_V_L100,_F())`, where `ABS_V_L100` is the CFSR absolute vorticity variable and `_F()` is a built-in function for computing the Coriolis parameter (defined by $f = 2\Omega \sin \phi$). In the northern hemisphere, we follow Thorncroft and Hodges (2001) and isolate tropical easterly wave features by requiring a drop of relative vorticity equal to $5 \times 10^{-5}$ s$^{-1}$. The command line used is as follows:

```
./DetectCyclonesUnstructured
```





```
   --in_data "$uvfile;$hfile" --out $outf
   --searchbymax "_DIFF(ABS_V_L100(0),_F())" --mergedist 2.0
   --closedcontourcmd "_DIFF(ABS_V_L100(0),_F()),-5.e-5,4,0"
   --outputcmd "ABS_V_L100(0),max,0;_DIFF(ABS_V_L100(0),_F()),max,0;
5       HGT_L1,max,0"
   --minlat -35.0 --maxlat 35.0
```

Tropical easterly wave paths are constructed using a maximum distance of 3° great-circle-distance between subsequent detections, a minimum path length equal to 8 sequential detections, no allowed gaps, and a distance of at least 10° between track start and endpoint. Northern (southern) hemisphere waves must also be present in the northern (southern) hemisphere for

10   at least 8 timesteps (2 days). The command line for northern hemisphere waves is as follows:

```
   ./StitchNodes
   --in pgbhnl.dcu_aew_nh_all.dat --out pgbhnl.dcu_aew_nh_stitch.dat
   --format "i,j,lon,lat,relv,zs"
   --range 3.0 --minlength 8 --maxgap 0
15   --min_endpoint_dist 10.0
   --threshold "lat,<=,25.0,8;lat,>=,0.0,8"
```

An analogous procedure is applied in the southern hemisphere, except searching on minima in the relative vorticity field and limiting the latitudinal range in `StitchNodes` to [25S,0] for at least 8 timesteps. Counts of total (northern hemisphere plus southern hemisphere) tropical easterly waves within each 2° grid volume are given in Figure 5, showing heavy wave

20   activity throughout the Atlantic and Pacific basins. These results are very similar to other reported easterly wave densities, such as Belanger et al. (2014) and Thorncroft and Hodges (2001), except for (a) the substantially enhanced tropical easterly wave count reported over eastern Africa (which could be eliminated by filtering over topography) and (b) essentially no observed wave activity off of the western coast of South America. Nonetheless, it is well known that easterly wave climatology varies strongly across reanalysis datasets and exhibits sensitivity to the choice of tracking scheme (Hodges et al., 2003).

25   ### 3.4 Tropical cyclones in a simulation with VR-CAM

For our final example, we assess the differences in tropical cyclone character obtained from native and regridded datasets. Using the variable-resolution spectral element option in the Community Atmosphere Model (VR-CAM-SE, Neale et al. (2012); Zarzycki et al. (2014b)) to refine the northern hemisphere to 0.25° resolution, a simulation of a hurricane season (June - November) has been performed. With the high-order spectral element dynamical core used to solve the fluid equations in the

30   atmosphere, VR-CAM-SE has been demonstrated to be effective in simulating tropical cyclone-like features (Zarzycki and Jablonowski, 2014; Zarzycki et al., 2014a; Zarzycki and Jablonowski, 2015). Since VR-CAM-SE uses an unstructured mesh with degrees of freedom stored at spectral element Gauss-Lobatto (GL) nodes, data is typically analyzed only after being regridded to a regular latitude-longitude mesh of approximately equal resolution. The regridding procedure has the potential to clip local extrema and smear out grid-scale features.





For this example, we use the high-order regridding package TempestRemap (Ullrich and Taylor, 2015; Ullrich et al., 2016) for remapping the native spectral element output to a regular latitude-longitude grid with 0.25° grid spacing. For purposes of determining connectivity on the variable-resolution spectral element mesh, we connect GL nodes along the coordinate axis of each quadrilateral element (see Figure 6). DetectCyclonesUnstructured is then applied to both the native grid data and the

5 regridded data on the regular latitude-longitude mesh (using the configuration specified in section 3.1) and tropical cyclones are categorized (color-coded) by maximum surface wind speed as defined by the Saffir-Simpson scale (Simpson, 1974), such that orange and red trajectories represent the strongest classifications of storms. The results of this analysis are depicted in Figure 7. As expected, the native grid output produces essentially identical tracks, but an increase in tropical cyclone intensity in some cases (with some tropical cyclones dropping down by a full category as a consequence of the remapping procedure

10 and discrete nature of binning storm strength).

## 4 Conclusions

Automated pointwise feature trackers have been frequently and successfully employed over the past several decades to extract useful information from large climate datasets. With spatial and temporal resolution increasing rapidly in response to enhanced computational resources, climate datasets have grown increasingly unwieldy and so there has been a growing need for such

large dataset processing tools. This paper has outlined a framework for pointwise feature tracking (TempestExtremes) that exposes a suite of generalized kernels drawn from the literature on trackers of the past several decades. This framework is sufficiently robust to be applicable to many climate and weather datasets, including data on unstructured grids. We expect such a framework would be useful for isolating uncertainties that emerge from particular parameter choices in tracking schemes, or to compute optimal threshold values for detecting pointwise features in, e.g. reanalysis data. Future development plans in Tem-

pestExtremes include the construction of analogous kernels for tracking areal features (blobs), such as clouds or atmospheric rivers.

## Code Availability

The open-source software described in this manuscript has been released as part of the TempestExtremes software package, and is available for use under the Lesser GNU Public License (LGPL). All software can be obtained from GitHub at:

```
https://github.com/ClimateGlobalChange/tempestextremes
```

## Appendix A: A Review of Extratropical Cyclone Tracking Algorithms

This appendix reviews the existing literature on extratropical cyclone tracking, one of the earliest and most common instances of both manual and automated feature tracking. Manual counts of cyclones were performed by Petterssen (1956) in the Northern hemisphere from 1899-1939, and latter binned by Klein (1957) to determine the spatial distribution of such storms. These

30 techniques were later refined by Whittaker and Horn (1982) by accounting for cyclone trajectories. A similar survey in the



Southern hemisphere was performed by Taljaard (1967) for July 1957 - December 1958. Manual tracking and characterization of cyclones was also performed by Akyildiz (1985) using ECMWF forecast data for the 1981/82 winter.

One of the first automated detection and tracking for extratropical cyclones was developed by Williamson (1981) using nonlinear optimization to fit cyclonic profiles to anomalies in the 500-mb geopotential height field. Storms were then tracked over a short forecast period using the best fit to the cyclone's centerpoint. Counts of cyclones neglecting the cyclone trajectory were automatically generated from climate model output for both hemispheres by Lambert (1988) using local minima in 1000-hPa geopotential height. This method had some shortcomings, including mischaracterization of local lows due to Gibbs' ringing and topographically-driven lows. To overcome these problems, Alpert et al. (1990) proposed an additional minimum threshold on the local pressure gradient. Similarly, Le Treut and Kalnay (1990) detected cyclones in ECMWF pressure data using a local minima in the sea-level pressure that must also be 4 hPa below the average sea-level pressure of neighboring grid points, and must persist for three successive 6- or 12- hour intervals. Murray and Simmonds (1991) extracted low pressure centers from interpolated GCM data using local optimization, based on earlier work in Rice (1982). These original papers primarily sought minimum in the SLP field or looked for maxima in the Laplacian of the SLP field.

Several modern extratropical cyclone detection algorithms remain in use, having built on this earlier work. Short descriptions of many of these schemes are given here. Some of these algorithms use the notion of a local neighborhood or periphery, as defined in Figure 8.

- Serreze et al. (1993); Serreze (1995): Assessed $\sim$ 381-400 km arctic dataset for extratopical cyclone behavior. Search on SLP for local minima at least 2 hPa higher than neighbors. Tracking is performed with a maximum search distance of 1400km per 12 hour period.

- Sinclair (1994, 1997): Assessed 2.5° ECMWF data over the southern hemisphere. Search on local minima in the 1000 hPa geostrophic vorticity field (computed from the Laplacian of the 1000 hPa geopotential), adjusted for topography and presence of heat lows (see paper for details), satisfying $\zeta_g < -2 \times 10^{-5} \mathrm{s}^{-1}$.

- Blender et al. (1997): Assessed T106 ECMWF analyses ($\sim$ 125 km). Search on local minima in the 1000 hPa geopotential height field. Require a positive mean gradient in the 1000 hPa geopotential height field in a $1000 \times 1000$ km$^2$ area around each candidate. Tracking is performed using nearest-neighbor search with a maximum displacement velocity of 80 km/h, eliminating cyclones with tracks shorter than 3 days.

- Lionello et al. (2002): Assessed a T106 ($\sim$ 125 km) ECHAM-4 dataset. Search on local minima in the SLP field. Tracking requires using previous cyclone velocity to delineate a prediction region, and tracks are discarded if they do not continue into the prediction region.

- Zolina and Gulev (2002): Assessed T106 ($\sim$ 125 km) and T42 ($\sim$ 300 km) datasets. Search on local minima in the SLP field.

- Hoskins and Hodges (2002); Catto et al. (2009); Dacre et al. (2012): Assessed various reanalysis and climate datasets with wavenumber $\leq$ 5 removed in all fields. Relative vorticity field is spectrally-truncated to T42. Search on 850 hPa





relative vorticity maxima. Trajectories are computed by searching for nearest neighbors and smoothed by minimizing a cost function. Cyclones must persist for 4 days.

– Pinto et al. (2005): Assessed T42 ($\sim$ 300 km) regridded NCEP reanalysis, regridded onto a 0.75° grid by cubic spline interpolation. Search on local minima in pressure field with maxima of quasi-geostrophic relative vorticity, computed from the Laplacian of pressure, within 1200km. Cyclones over topography above 1500m are removed. Require quasi-geostrophic relative vorticity $> 0.1$hPa/(°lat) and retain only the strongest detections within 3°. Cyclone tracking requires a prediction velocity and search following Murray and Simmonds (1991).

– Benestad and Chen (2006): Assessed 2.5° ERA40 data. Search uses multiple least-squares regression to estimate the values of the coefficients of a Fourier approximation followed by 1D search in north-south and east-west directions (in effect smoothing the SLP field).

– Simmonds et al. (2008): Assessed several 2.5° datasets over the arctic. Search on local minima in the Laplacian of pressure, rejecting cyclones over topography above 1000m and requiring the presence of a nearby pressure minima. Identified lows must satisfy a Laplacian with value $> 0.2$hPa/(°lat)$^2$ over a radius of 2°. Tracking uses a probability estimate using a predicted position.

## Appendix B: A Review of Tropical Cyclone Tracking Algorithms

More recently, and as higher resolution climate data has become available, extratropical cyclone tracking techniques have been modified in order to support tropical cyclone tracking. To eliminate "false positives" associated with extratropical cyclones and weak cyclonic depressions, many schemes require that the candidate be associated with a nearby warm core and be associated with a minimum threshold on surface winds for at least 1-3 days. The definition of a "warm core" varies between modeling centers, including such options as air temperature anomaly on pressure surfaces (Vitart et al., 1997; Zhao et al., 2009; Murakami et al., 2012), geopotential thickness (Tsutsui and Kasahara, 1996) and decay of vorticity with height (Bengtsson et al., 2007a; Strachan et al., 2013). Additional filtering of candidate storms over topography or within a specified latitudinal range may be required. To better match observations, additional geographical, model or feature-dependent criteria may be applied (Camargo and Zebiak, 2002; Walsh et al., 2007; Murakami and Sugi, 2010a; Murakami et al., 2012). It is widely acknowledged that weaker tropical storms are difficult to track, and the observational record of these less-intense, short-lived storms is questionable (Landsea et al., 2010).

A tabulated overview of the thresholds utilized by many of these schemes can be found in Walsh et al. (2007), along with several proposed guidelines on detection schemes. We extend this tabulation with the following short descriptions of many published schemes.

– Bengtsson et al. (1982): Assessed one year of $\sim$ 200 km forecast model output. Search latitude $< 30°$ for collocated 850hPa wind $> 25$ m/s and 850hPa relative vorticity maxima $> 7 \times 10^{-5}$s$^{-1}$ in 7.5°x7.5° area.



- Broccoli and Manabe (1990): Assessed a R15 ($\sim$ 600 km) and R30 ($\sim$ 300 km) dataset. Search on PSL with 1.5 hPa local min (R15) or 0.75 hPa local min (R30), with local surface wind velocity > 17 m/s, latitude < 30°. Cyclones are tracked over a range of 1200 km / day.

- Wu and Lau (1992): Assessed a 7.5° longitude × 4.5° latitude dataset. Search on minimum 1000hPa geopotential height, requiring positive 950hPa relative vorticity, negative 950hPa divergence, positive 500hPa vertical velocity, latitude < 40.5°, 200hPa minus 1000hPa layer thickness must be locally maximal and exceed by 60 m the average layer thickness within 1500km west to east, and 950hPa wind must be > 17.2 m/s locally. Cyclones are tracked over a range of 7.5° longitude or 9° latitude per day.

- Haarsma et al. (1993): Assessed a $\sim$ 300 km dataset. Search on local minimum PSL and require 850hPa relative vorticity $> 3.5 \times 10^{-5}\mathrm{s}^{-1}$, and temperature anomaly at 250hPa $\Delta T250 > 0.5$K, at 500hPa $\Delta T500 > -0.5$K, and $\Delta T250 - \Delta T850 > -1.0$K, where the anomaly is computed against a $15° \times 15°$ spatial mean around the center of the storm. Cyclones are tracked for a minimum of 3 days.

- Bengtsson et al. (1995, 1996): Assessed a T106 ($\sim$ 125 km) dataset. Search on 850hPa relative vorticity $> 3.5 \times 10^{-5}\mathrm{s}^{-1}$. Require a 850hPa wind maximum > 15 m/s, local SLP minimum, and mean 850hPa wind > mean 300hPa wind within 7x7 grid points around candidate. Further require temperature anomaly sum $\Delta T700 + \Delta T500 + \Delta T300 > 3$ K and $\Delta T300 > \Delta T850$ where the anomaly is computed against a 7x7 gridpoint average centered on candidate. Cyclones are tracked for a minimum of 1.5 days.

- Tsutsui and Kasahara (1996): Assessed a T42 (2.8°, $\sim$ 300 km) dataset. Search on minima in 1000hPa geopotential height field, with at least an average drop of 20 m among neighboring points, and a further 20 m drop of average among neighboring points from periphery. Require average local 900hPa vorticity to be cyclonic, average local 900hPa divergence to be negative, average local 500hPa vertical velocity to be upward, 200hPa minus 1000hPa layer thickness maximum among neighbors is greater than any value in periphery, and average local 200hPa zonal wind velocity is less than 10 m/s or local points contain at least one point with easterly velocity. Require the latitude < 40°, topographic height underlying candidates should be < 400 m, one local point must have a 900hPa wind speed of at least 17.2 m/s, and one local point must exceed 100 mm/d over at least one day. Cyclones are tracked for a minimum of 2 days.

- Vitart et al. (1997, 1999); Vitart et al. (2001); Vitart et al. (2003): Assessed a T42 (2.8°, $\sim$ 300 km) dataset. Search on 850hPa relative vorticity maxima $> 3.5 \times 10^{-5}\mathrm{s}^{-1}$ with a nearby PSL minimum. Must possess a warm core within 2° latitude defined as a local average 500hPa to 200hPa temperature maximum with a decrease of 0.5 K in all directions within 8°. Must possess a local maximum in 200hPa - 1000hPa thickness with a decrease of 50m in all directions within 8°. When tracking, the minimum distance between storms is 800 km / day, tracks must last at least 2 days and the maximum wind velocity within 8° of the storm center must be 17 m/s for at least 2 (not necessarily consecutive) days.

- Walsh (1997); Walsh and Watterson (1997); Walsh and Katzfey (2000): Assessed a 125 km regional climate dataset over Australia. Required 850 hPa relative vorticity $> 2.0 \times 10^{-5}\mathrm{s}^{-1}$, temperature anomaly sum $\Delta T700 + \Delta T500 + \Delta T300 >$



0 K and $\Delta T300 > \Delta T850$, with anomaly computed against the mean over a region 2 grid points north/south and 13 grid points east/west. Further require 10m surface wind $> 10$ m/s and 850 hPa tangential wind speed $> 300$ hPa tangential wind speed. Cyclones are tracked for a minimum of 2 days.

- Krishnamurti et al. (1998): Assessed a T42 ($\sim 300$ km) climate dataset. Similar to Bengtsson et al. (1995, 1996), except using a 4x4 grid point region for 850hPa wind maximum, SLP minimum and temperature mean. Cyclones are tracked for at least 1 day.

- Nguyen and Walsh (2001): Similar to Walsh and Watterson (1997), assessed a 125 km regional dataset over Australia. Vorticity requirement was changed to 850 hPa relative vorticity $> 1.0 \times 10^{-5}\mathrm{s}^{-1}$ with PSL minimum within 250km. Also required mean wind speed in 500km $\times$ 500km region at 850 hPa was larger than at 300 hPa and a mean tangential wind speed within a radius of $1°$ and $2.5°$ greater than 5 m/s. Cyclones were tracked for a minimum of 1 day, with relaxed criteria after this time.

- Sugi et al. (2002): Assessed a T106 ($\sim 125$ km) climate dataset. Tracking criteria similar to Bengtsson et al. (1995). Search is conducted for local PSL minima that are at least $< 1020$ hPa. Cyclones are tracked for a minimum of 2 days.

- Camargo and Zebiak (2002): Assessed a T42 ($\sim 300$ km) climate dataset. Similar to Bengtsson et al. (1995, 1996), except basin-specific thresholds are applied for 850hPa relative vorticity, 850hPa wind speed, and temperature anomaly sum $\Delta T700 + \Delta T500 + \Delta T300$. Thresholds are determined by sampling the tails of probability density functions for relevant variables in each ocean basin. Following detection stage, apply relaxed 850hPa relative vorticity threshold ($> 1.5 \times 10^{-5}\mathrm{s}^{-1}$) to area 3x3 grid points around prior detections to construct trajectories. Cyclones are tracked for at least 2 (1.5) days in daily (6-hourly) output.

- Tsutsui (2002): Assessed a T42 ($2.8°$, $\sim 300$ km) dataset. Search is performed similar to Tsutsui and Kasahara (1996), but with a simplified criteria. PSL is required to be less than the local average minus 2 hPa, and local average must be less than periphery average minus 2hPa. Layer thickness between 200hPa and 700hPa, denoted by $Z$, must satisfy $Z_0 + \max(Z_{\pm 1\Delta}) > 2\max(Z_{\pm 2\Delta})$, where $Z_{\pm 1\Delta}$ denotes immediate neighbors and $Z_{\pm 2\Delta}$ denotes the periphery.

- Cheung and Elsberry (2002); Halperin et al. (2013): Similar to Walsh (1997). Assessed tropical cyclogenesis in weather forecast models. Required a gridpoint maximum in 850 hPa relative vorticity larger than all surrounding gridpoints within $4°$ and either a local maximum in 200-500 hPa average temperature or 200-1000 hPa geopotential thickness that was offset by no more than $2°$ from associated PSL center. In Halperin et al. (2013), TCs must persist for at least 24 hours and have a 925 hPa horizontal wind speed greater than a model-specific threshold within $5°$ of PSL center.

- Walsh et al. (2004): Assessed a 30 km dataset using a similar tracking strategy to Nguyen and Walsh (2001). The temperature anomaly was computed against a 1200km longitude $\times$ 400km latitude region, and the mean wind speed was computed over a 800km $\times$ 800km region around the storm. Further required that V10 $\geq 17$ m/s near storm.





– McDonald et al. (2005): Assessed a 2.5° latitude by 3.75° longitude dataset. Search on local maxima of 850hPa relative vorticity with magnitude greater than $5 \times 10^{-5} \mathrm{s}^{-1}$ with initial latitude $< 30°$. Temperature anomalies must satisfy $\Delta T300 > 0$ along the track, $\Delta T300 > 0.5$ K for any two points along the track and $\Delta T300 > \Delta T850$ for any two points along the track, where the anomaly was computed against a $15° \times 15°$ mean. Cyclones are tracked for a minimum of 2 days.

– Chauvin et al. (2006): Assessed a T319 ($\sim 50$ km) climate timeslice simulation. Search on local minimum PSL and require 850hPa relative vorticity $> 1.4 \times 10^{-4} \mathrm{s}^{-1}$, 850hPa wind $> 15$ m/s, mean 700-300 hPa temperature anomaly $\Delta \overline{T700 - T300} > 3$K, $\Delta T300 > \Delta T850$, and 850 hPa wind $> 300$ hPa wind. Anomalies are computed against environmental values 500km from the cyclone center. Applies similar relaxation technique as Camargo and Zebiak (2002) to eliminate split trajectories. Cyclones are tracked for a minimum of 1.5 days.

– Oouchi et al. (2006): Assessed a 20 km dataset using a similar technique to Sugi et al. (2002). PSL at storm center must be 2 hPa lower than mean over 7x7 grid box and require that storm center latitude $< 45°$ with an initial position $< 30°$. Near the storm require relative vorticity at 850 hPa must be $> 3.5 \times 10^{-5} \mathrm{s}^{-1}$, maximum wind speed at 850hPa must be $> 15$ m/s, and the maximum wind speed at 850hPa is larger than at 300hPa. Further require temperature anomaly sum $\Delta T700 + \Delta T500 + \Delta T300 > 2$ K near storm. Cyclones are tracked for at least 1.5 days.

– Bengtsson et al. (2007b): Assessed T63, T213 and T319 datasets. Required 850hPa relative vorticity minus 250hPa relative vorticity exceed $6 \times 10^{-5} \mathrm{s}^{-1}$, that 850 hPa relative vorticity $> 6 \times 10^{-5} \mathrm{s}^{-1}$ and that relative vorticity be positive for all levels between 850 hPa and 250 hPa. Only northern hemisphere cyclones were preserved (latitude $< 60°$). Cyclones are tracked for a minimum of 1 day.

– Knutson et al. (2007); Zhao et al. (2009): Similar to Vitart et al. (1997, 2003), assessed a $\sim 50$ km dataset. Search on absolute 850hPa relative vorticity maxima $> 1.6 \times 10^{-4} \mathrm{s}^{-1}$ within $6° \times 6°$ areas with a local minimum in SLP within $2°$ of the detection. Further require a maximum in average temperature between 300 hPa and 500 hPa within $2°$ of the detection that is 1K warmer than the local mean. Cyclones are tracked for a minimum of 3 days, with a maximum search radius of 400km per 6 hours, and requiring that at least 3 days have a maximum surface wind speed greater than 17 m/s.

– Murakami and Sugi (2010b): Assessed four datasets with resolutions from TL95 ($\sim 180$ km) to TL959 ($\sim 20$ km). Cyclone identification similar to Oouchi et al. (2006) with a resolution-dependent relative vorticity criteria.

– Caron et al. (2011, 2013): Assessed 0.3° ($\sim 35$ km) climate model. Search for SLP minimum (minima merged if within $2°$), 850 hPa relative vorticity $> 4.0 \times 10^{-5} \mathrm{s}^{-1}$, temperature anomalies $\Delta T500 > 1$K and $\Delta T250 > 0$K (calculated relative to $5°$ radial mean around TC center), 850 hPa relative vorticity $> 250$ hPa relative vorticity, and resolution-specific surface wind threshold as in Walsh et al. (2007). Cyclones are tracked for a minimum of 24 hours. Tracks with a relaxed set of thresholds are calculated in parallel and applied to the main tracking to minimize broken trajectories, similar to Camargo and Zebiak (2002).



- Murakami et al. (2012): Assessed four datasets with resolutions from 20 km to 60 km. Cyclone identification similar to Oouchi et al. (2006) with a resolution-dependent relative vorticity and temperature anomaly criteria. Temperature anomalies are computed against a $10° \times 10°$ grid box. Additional filtering is applied in the North Indian Ocean requiring maximum wind speed be within 100-200km of storm center. Tracking incorporates a maximum gap size of 1 (a single time step failure).

- Au-Yeung and Chan (2012); Huang and Chan (2014): Assessed a $\sim 60$km dataset. Required 850 hPa relative vorticity $> 4.5 \times 10^{-4}$s$^{-1}$ and 300 hPa temperature anomaly of 1°K defined relative to 15° radius around vorticity center. Cyclones are tracked for a minimum of 2 days and must originate over the ocean.

- Tory et al. (2013c, b, a): Assessed datasets at 1°x1° resolution. Requires Okubo-Weiss-Zeta parameter values $> 50 \times 10^{-6}$s$^{-1}$ and $> 40 \times 10^{-6}$s$^{-1}$ at 850 and 500 hPa, relative humidity $> 70\%$ and $> 50\%$ at 950 and 700 hPa, and 850-200 hPa vertical wind shear $< 25$ m/s. Storm must persist for 2 days (48 hours).

- Strachan et al. (2013): Assessed datasets from $\sim 60$km to $\sim 270$ km. At T63 resolution, required that 850hPa relative vorticity attain a threshold of $> 6 \times 10^{-5}$s$^{-1}$, and relative vorticity at 500 hPa and 200 hPa be positive. Further required that the relative vorticity difference between 850 hPa and 200 hPa $> 6 \times 10^{-5}$s$^{-1}$. Cyclones are tracked for a minimum of 1 day.

- Horn et al. (2014): Assessed selected datasets from U.S. Climate Variability and Predictability Research Program (CLI-VAR) Hurricane Working Group (HWG) (Walsh et al., 2015) ($\sim 60$km to $\sim 110$ km). Similar to Walsh et al. (2004), except resolution-dependent value for surface winds are applied based on Walsh et al. (2007) and tropical cyclones are differentiated from extratropical storms by enforcing tropical cyclone genesis points to be equatorward of extratropical ridges in both hemispheres.

- Zarzycki and Jablonowski (2014): Assessed a $\sim 28$km dataset. Search on absolute 850hPa relative vorticity maxima $> 1.0 \times 10^{-4}$s$^{-1}$ with latitude $< 45°$ and SLP minimum within 4°. Require a local maximum 500-200 hPa average temperature within 2° of storm center which decreases by at least 0.8 K at a radius of 5° in all directions. Cyclones are tracked for a minimum of 2 days, with a maximum search radius of 400km per 6 hours, and requiring that at least 2 days have a maximum surface wind speed is greater than 17 m/s within 4° of the candidate. Tracking allows for a maximum gap size of 1 (cyclone lysis and subsequent genesis occuring within 400km and 12 hours of each other are merged).

- Reed and Chavas (2015): Tracked tropical cyclones on a planet in radiative-convective equilibrium. Search on SLP minima followed by a closed contour criteria that requires a pressure increase of at least 4 hPa in all directions within 5 degrees (great-circle distance) using an early release of TempestExtremes.

- Bosler et al. (2016): Assessed a $\sim 14$km dataset. Similar to Knutson et al. (2007); Zhao et al. (2009) except uses great-circle-distances for spatial calculations instead of grid point search.





- Harris et al. (2016): Assessed multiple climate datasets with regional resolution ranging from $\sim$ 75km to $\sim$ 10km. Searches on minimum smoothed SLP (no greater than 1013hPa) with 2hPa closed contour not encircling another minimum. Storms required to have $2°$K closed contour around 300-500 hPa temperature maximum within 500km and an 850 hPa relative vorticity $> 1.5 \times 10^{-4}\text{s}^{-1}$. Cyclones are tracked for a minimum of 3 days, with a maximum search radius

  of 750km per 6 hours, and require at least 1.5 consecutive days of a maximum surface wind speed greater than 17.5 m/s, following Chen and Lin (2011)

## Appendix C: A (Short) Review of Tropical Easterly Wave Tracking Algorithms

Tropical easterly waves are featured more sparsely within the literature, but are nonetheless an important pointwise feature in climate datasets. Pointwise tracking is complementary to statistical techniques which typically examine the variability, for

instance, in the African Easterly Jet (AEJ) (*i.e.*, Ceron and Gueremy (1999)). The first manual study that identified and tracked African easterly wave was performed by Reed et al. (1988) using positive relative vorticity anomalies. This strategy was also applied by Thorncroft and Hodges (2001), Hodges et al. (2003) and Serra et al. (2010). Other studies have used curvature vorticity anomalies Agudelo et al. (2011); Belanger et al. (2014); Bain et al. (2014); Brammer and Thorncroft (2015) and streamfunctions (Berry et al., 2007).

## Appendix D: Software Documentation: DetectCyclonesUnstructured

This section contains the software documentation for the executable `DetectCyclonesUnstructured` from the TempestExtremes package. The software is provided for use within a command-line shell, such as bash.

```
Usage: DetectCyclonesUnstructured 
Parameters:  --in_data <string> [""]
```

```
  --in_data_list <string> [""]
  --in_connect <string> [""]
  --out <string> [""]
  --out_file_list <string> [""]
```
25
```
  --searchbymin <string> [""] (default PSL)
  --searchbymax <string> [""]
  --minlon <double> [0.000000] (degrees)
  --maxlon <double> [0.000000] (degrees)
  --minlat <double> [0.000000] (degrees)
```
30
```
  --maxlat <double> [0.000000] (degrees)
  --minabslat <double> [0.000000] (degrees)
  --topofile <string> [""]
```





```
     --maxtopoht <double> [0.000000] (m)

     --mergedist <double> [0.000000] (degrees)

     --closedcontourcmd <string> [""] [var,delta,dist,minmaxdist;...]

     --noclosedcontourcmd <string> [""] [var,delta,dist,minmaxdist;...]

--thresholdcmd <string> [""] [var,op,value,dist;...]

     --outputcmd <string> [""] [var,op,dist;...]

     --timestride <integer> [1]

     --regional <bool> [false]

     --out_header <bool> [false]

--verbosity <integer> [0]

     --in_data <string>
```

A list of input datafiles in NetCDF format, separated by semi-colons.

```
     --in_data_list <string>
```

A file containing the --in_data argument for a sequence of processing operations (one per line).

```
--in_connect <string>
```

A connectivity file, which uses a vertex list to describe the graph structure of the input grid. This parameter is not required if the data is on a latitude-longitude grid.

```
     --out <string>
```

The output file containing the filtered list of candidates in plain text format.

```
--out_file_list <string>
```

A file containing the --out argument for a sequence of processing operations (one per line).

```
     --searchbymin <string>
```

The input variable to use for initially selecting candidate points (defined as local minima). By default this is "PSL", representing detection of surface pressure minima. Only one of searchbymin and searchbymax may be set.

```
--searchbymax <string>
```

The input variable to use for initially selecting candidate points (defined as local maxima). Only one of searchbymin and searchbymax may be set.

```
     --minlon <double>
```

The minimum longitude for candidate points.

```
--maxlon <double>
```

The maximum longitude for candidate points.





`--minlat <double>`

The minimum latitude for candidate points.

`--maxlat <double>`

The maximum latitude for candidate points.

`--minabslat <double>`

The minimum absolute latitude for candidate points.

`--mergedist <double>`

Merge candidate points with distance (in degrees) shorter than the specified value. Among two candidates within the merge distance, only the candidate with lowest `searchbymin` or highest `searchbymax` value will be retained.

`--closedcontourcmd <cmd1>;<cmd2>;...` Eliminate candidates if they do not have a closed contour. Closed contour commands are separated by a semi-colon. Each closed contour command takes the form `var,delta,dist,minmaxdist`. These arguments are as follows.

      `var <variable>` The variable used for the contour search.

      `dist <double>` The great-circle distance (in degrees) from the pivot within which the closed contour criteria

must be satisfied.

      `delta <double>` The amount by which the field must change from the pivot value. If positive (negative) the field must increase (decrease) by this value along the contour.

      `minmaxdist <double>` The distance away from the candidate to search for the minima/maxima. If `delta` is positive (negative), the pivot is a local minimum (maximum).

`--noclosedcontourcmd <cmd1>;<cmd2>;...`

As `closedcontourcmd`, except eliminates candidates if a closed contour is present.

`--thresholdcmd <cmd1>;<cmd2>;...` Eliminate candidates that do not satisfy a threshold criteria (there must exist a point within a given distance of the candidate that satisfies a given equality or inequality). Threshold commands are separated by a semi-colon. Each threshold command takes the form `var,op,value,dist`. These arguments are

as follows.

      `var <variable>` The variable used for the contour search.

      `op <string>` Operator that must be satisfied for threshold (options include >, >=, <, <=, =, !=).

      `value <double>` The value on the RHS of the comparison.

      `dist <double>` The great-circle-distance away from the candidate to search for a point that satisfies the thresh-

old (in degrees).





`--outputcmd <cmd1>;<cmd2>;...` Include additional columns in the output file. Output commands take the form `var,op,dist`. These arguments are as follows.

> `var <variable>` The variable used for the contour search.

> `op <string>` Operator that is applied over all points within the specified distance of the candidate (options include `max`, `min`, `avg`, `maxdist`, `mindist`).

> `dist <double>` The great-circle-distance away from the candidate wherein the operator is applied (in degrees).

`--timestride <integer>`

Only examine discrete times at the given stride (by default 1).

`--regional`

When a latitude-longitude grid is employed, do not assume longitudinal boundaries to be periodic.

`--out_header`

Output a header describing the columns of the data file.

`--verbosity <integer>`

Set the verbosity level (default 0).

## D1 Variable Specification

Quantities of type `<variable>` include both NetCDF variables in the input file (for example, "Z850") and simple operations performed on those variables. By default it is assumed that NetCDF variables are specified in the `.nc` file as

$$\texttt{float Z850(time, lat, lon)} \quad \text{or} \quad \texttt{float Z850(time, ncol)}$$

for structured latitude-longitude grids and unstructured grids, respectively. If variables have no time variable, they have the related specification

$$\texttt{float Z850(lat, lon)} \quad \text{or} \quad \texttt{float Z850(ncol)}$$

If variables include an additional dimension, for instance,

$$\texttt{float Z(time, lev, lat, lon)} \quad \text{or} \quad \texttt{float Z(time, lev, ncol)}$$

they may be specified on the command-line as `Z(<lev>)`, where the integer index `<lev>` corresponds to the first dimension (or the dimension after `time`, if present).

Simple operators are also supported, including

`_ABS(<variable>)` Absolute value of a variable,

`_AVG(<variable>, <variable>)` Pointwise average of variables,





`_DIFF(<variable>, <variable>)` Pointwise difference of variables,

`_F()` Coriolis parameter,

`_MEAN(<variable>, <distance>)` Spatial mean over a given radius,

`_PLUS(<variable>, <variable>)` Pointwise sum of variables,

`_VECMAG(<variable>, <variable>)` 2-component vector magnitude.

For instance, the following are valid examples of `<variable>` type,

$$\texttt{\_MEAN(PSL,2.0),} \quad \texttt{\_VECMAG(U850, V850)} \quad \text{and} \quad \texttt{\_DIFF(U(3),U(5)).}$$

## D2   MPI Support

The `DetectCyclonesUnstructured` executable supports parallelization via MPI when the `--in_data_list` argu-
ment is specified. When enabled, the parallelization procedure simply distributes the processing operations evenly among
available MPI threads.

## Appendix E:  Software Documentation: StitchNodes

This section contains the software documentation for the executable `StitchNodes` from the TempestExtremes package.

```
Usage: StitchNodes 
```
```
Parameters:
  --in <string> [""]
  --out <string> [""]
  --format <string> ["no,i,j,lon,lat"]
  --range <double> [5.000000] (degrees)
```
```
  --minlength <integer> [3]
  --min_endpoint_dist <double> [0.000000] (degrees)
  --min_path_dist <double> [0.000000] (degrees)
  --maxgap <integer> [0]
  --threshold <string> [""] [col,op,value,count;...]
```
```
  --timestride <integer> [1]
  --out_format <string> ["std"] (std|visit)

  --in <string>
```
The input file (a list of candidates from DetectCyclonesUnstructured).

```
  --out <string>
```
The output file containing the filtered list of candidates in plain text format.





`--format <string>`

The structure of the columns of the input file.

`--range <double>`

The maximum distance between candidates along a path.

`--minlength <integer>`

The minimum length of a path (in terms of number of discrete times).

`--min_endpoint_dist <double>`

The minimum great-circle distance between the first candidate on a path and the last candidate (in degrees).

`--min_path_dist <double>`

The minimum path length, defined as the sum of all great-circle distances between candidate nodes (in degrees).

`--maxgap <integer>`

The largest gap (missing candidate nodes) along the path (in discrete time points).

`--threshold <cmd1>;<cmd2>;...`

Eliminate paths that do not satisfy a threshold criteria (a specified number of candidates along path must satisfy an equality or inequality). Threshold commands are separated by a semi-colon. Each threshold command takes the form `col,op,value,count`. These arguments are as follows.

`col <integer>` The column in the input file to use in the threshold criteria.

`op <string>` Operator used for comparison of column value (options include >, >=, <, <=, =, !=).

`value <double>` The value on the right-hand-side of the operator.

`count <integer>` The minimum number of candidates along the path that must satisfy this criteria.

`--timestride <integer>`

Only examine discrete times at the given stride (by default 1).

*Acknowledgements.* This work has been supported by NASA award NNX16AG62G "TempestExtremes: Indicators of change in the characteristics of extreme weather." The authors would like to thank Dr. Kevin Reed for his efforts ensuring the quality of the software package.



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

**Algorithm 1** Compute the spatial mean value of a field `G` over a region of radius `dist` using graph search on an unstructured grid.

```
field F = mean(field G, dist)
  for each node p
    total_area = 0
    F[p] = 0
    visited = []
    tovisit = [p]
    while tovisit is not empty
      q = remove node from tovisit
      add q to visited
      F[p] = F[p] + G[q] * area[q]
      total_area = total_area + area[q]
      for each neighbor s of q
        if (gcd(p,s) < dist) and (s is not in visited) then
          add s to tovisit
    F[p] = F[p] / total_area
```




**Algorithm 2** Locate the set of all nodes `P` that are local minima for a field `G` (for instance, SLP) defined on an unstructured grid. The procedure for locating maxima is analogous.

```
set P = find_all_minima(field G)
  for each node f
    is_minima[f] = true
    for each neighbor node v of f
      if G[v] < G[f] then
        is_minima[f] = false
    if is_minima[f] then
      insert f into P
```

**Algorithm 3** Given a field `G` defined on an unstructured grid and a set of candidate points `P`, remove candidate minima that are within a distance `dist` of a more extreme minimum, and return the new candidate set `Q`.

```
set Q = merge_candidates_minima(field G, set P, dist)
  K = build_kd_tree(P)
  for each candidate p in P
    retain_p = true
    N = kd_tree_all_neighbors(K, p, dist)
    for all q in N
      if (G[q] < G[p]) then retain_p = false
    if retain_p then insert p into Q
```





---

**Algorithm 4** Find the node `pmax` containing the maximal value of the field `G` within a distance `maxdist` of the node `p`. An analogous procedure `find_min_near` is provided for locating nodes containing minimal values of the field.

---

```
node pmax = find_max_near(node p, field G, maxdist)
  set visited = {}
  set tovisit = {p}
  pmax = p
  while tovisit is not empty
    q = remove node from tovisit
    if (q in visited) then continue
    add q to visited
    if (gcdist(p,q) > maxdist) then continue
    if (G[q] > G[pmax]) then pmax = q
```

---

**Algorithm 5** Determine if there is a closed contour in field `G` of magnitude `thresh` around the point `p0`, defined by `p0 = find_max_near(p, G, maxdist)`, within distance `dist`. That is, along all paths away from `p0`, the field `G` must drop by at least `thresh` within distance `dist`. The closed contour criteria is depicted in Figure 2. An analogous procedure is defined for closed contours around minima.

---

```
closed_contour_max(point p, field G, dist, maxdist, thresh)
  p0 = find_max_near(p, G, maxdist)
  set visited = {}
  set tovisit = {p0}
  while tovisit is not empty
    q = remove point from tovisit
    if (q in visited) then continue
    add q to visited
    if (gcdist(p0,q) > dist) then return false
    if (G[p0] - G[q] < thresh) then
      add all neighbors of q to tovisit
  return true
```

---



---

**Algorithm 6** Determine if a candidate node `p` satisfies the requirement that there exists another node `p0` within distance `dist` of `p` with `G[p] > thresh`.

---

```
threshold_max(node p, field G, dist, thresh)
  p0 = find_max_near(p, G, dist)
  if (G[p0] < thresh) then
    return false
  else
    return true
```

---

**Algorithm 7** Determine all feature paths `S`, given array of candidate nodes `P[1..T]` and maximum great-circle distance between nodes at subsequent time levels `dist`.

---

```
path set S = stitch_nodes(set array P[1..T], dist, maxgap)
  for each time level t = 1..T
    K[t] = build_kd_tree(P[t])
  for each time level t = 1..T
    while P[t] is not empty
      initialize empty path s
      p = remove next candidate from P[t]
      add p into s
      gap = 0
      for time level u = t+1..T
        q = kd_tree_nearest_neighbor(K[u], p)
        if (q in P[u]) and (gcdist(p,q) < dist) then
          add q into s
          remove q from P[u]
          p = q
        else if (gap < maxgap) then
          gap = gap + 1
        else
          break
      add s into S
```

---





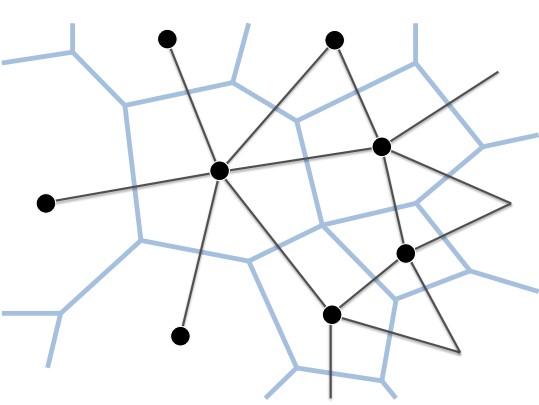

**Figure 1.** An example node graph describing an unstructured grid (blue lines), where nodes are co-located with volume centerpoint locations (solid circles) and edges connect adjacent volumes.





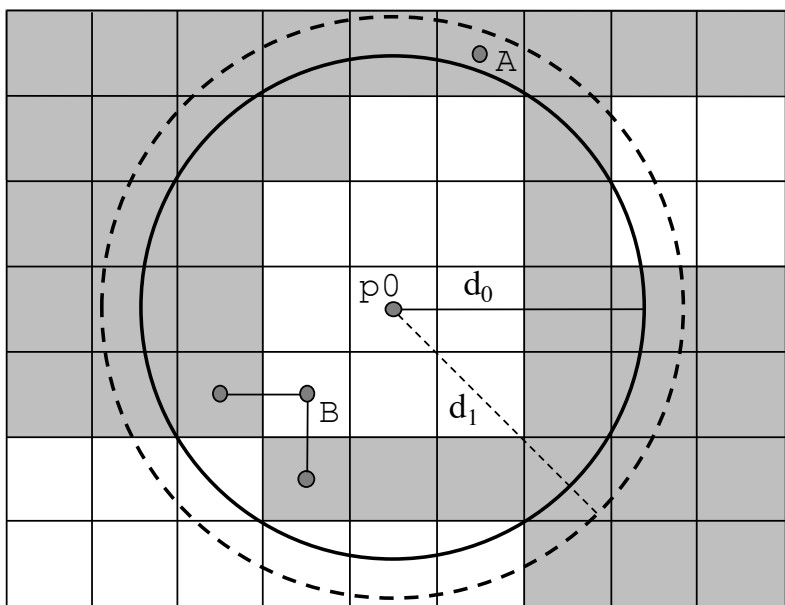

**Figure 2.** An illustration of the closed contour criteria. Nodes shaded in white (gray) satisfy (do not satisfy) the threshold of the field value at p0. Since only edge-neighbors are included, B constitutes a boundary to the interior of the closed contour. Because A lays outside the solid circle, the contour with distance $d_0$ is not a closed contour, whereas the dashed contour with distance $d_1$ does satisfy the closed contour criteria.

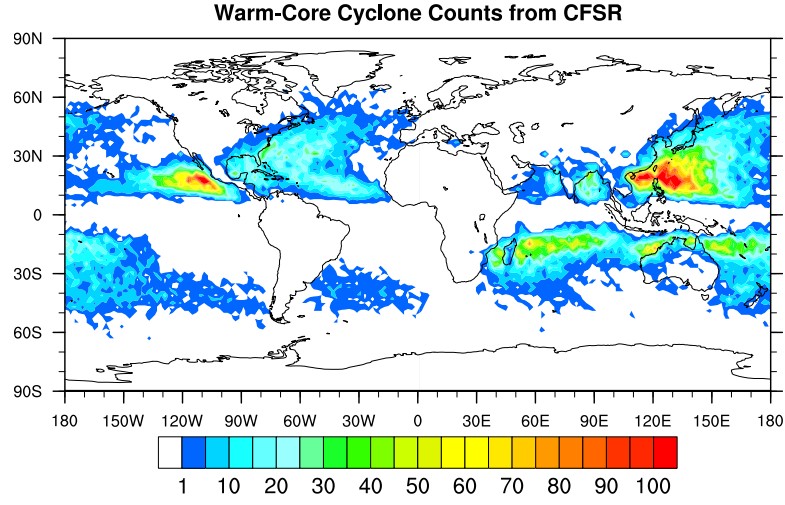

**Figure 3.** Tropical cyclone counts within each $2° \times 2°$ grid cell over the period 1979-2010 obtained using the procedure described in section 3.1.




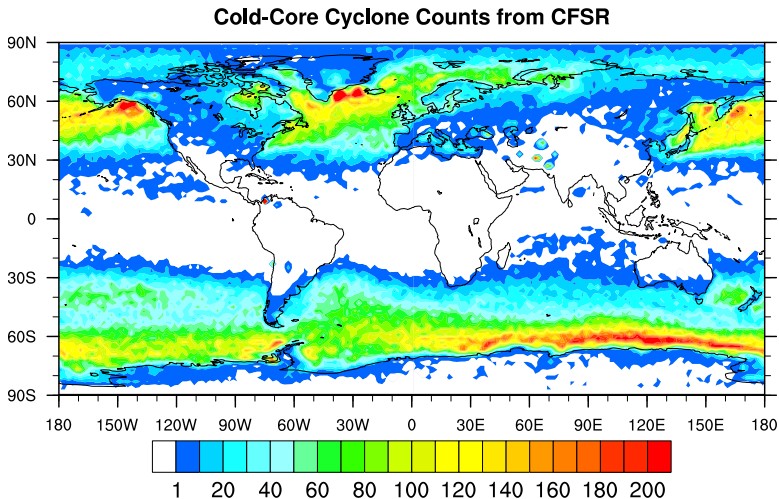

**Figure 4.** Extratropical cyclone counts within each $2° \times 2°$ grid cell over the period 1979-2010 obtained using the procedure described in section 3.2.

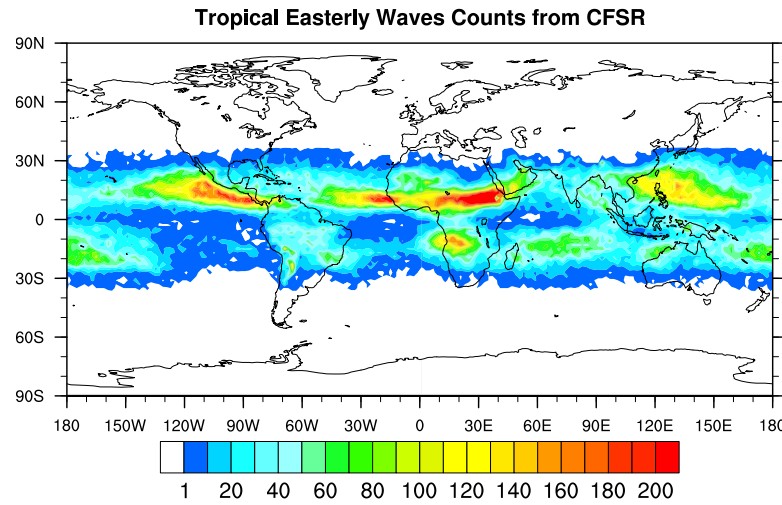

**Figure 5.** Tropical easterly wave counts within each $2° \times 2°$ grid cell over the period 1979-2010 obtained using the procedure described in section 3.3.




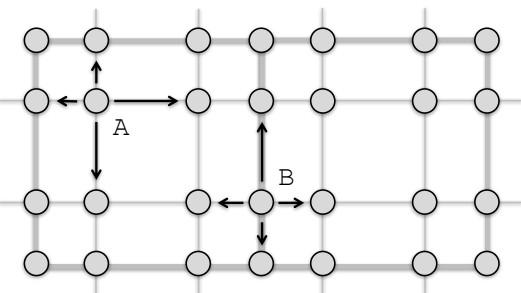

**Figure 6.** An illustration of how connectivity is defined in this work for nodes on a spectral element mesh. Arrows indicate connectivity for nodes A and B.

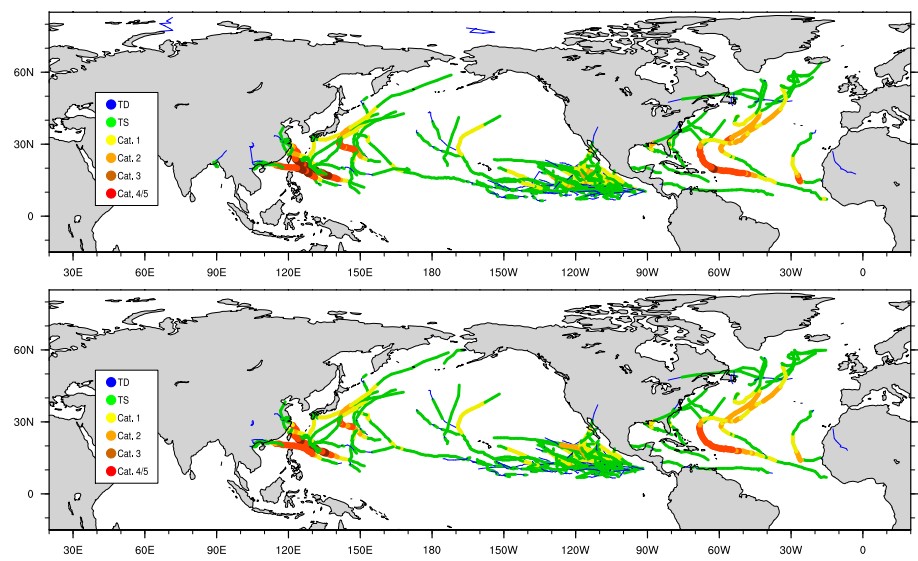

**Figure 7.** Tropical cyclone trajectories and associated intensities as obtained from the simulation of a single hurricane season in CAM 3.4 using (top) native spectral-element grid data and (bottom) data regridded to a regular latitude-longitude grid with 0.25° grid spacing.





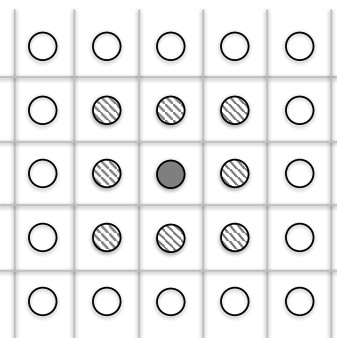

**Figure 8.** The local neighborhood of a central node (shaded) typically refers to the surrounding 8 nodes (diagonal hatching). The periphery (used by Tsutsui and Kasahara (1996)), refers to the set of nodes that surround the local neighborhood (unshaded nodes).