# Peer review of "TempestExtremes v1.0: A Framework for Scale-Insensitive Pointwise Feature Tracking on Unstructured Grids"

_Geoscientific Model Development, 2016_

## Referee Comment (RC1) · Anonymous Referee #1 · 27 Oct 2016

This is an excellent manuscript, very well written, in an important topic with many applications. I recommend the publication of the manuscript with minor revisions. My main suggestion is that the authors should expand the explanations of their method, as detailed below.

The issue of tracking features in models is an important one, and has important issues. It has many implications, as it hinders model intercomparisons, as well could have important implications in model projections. Therefore, having an open code that can be used in different models, with different resolutions and for multiple features is extremely attractive.

I like the solution of using a great-circle-distance for dealing with models with different

resolutions. It would be good for the authors to expand on this topic a little bit, maybe showing an idealized example of with plots of the great-circle distance in models of two different resolutions, to make the solution clearer for the reader. Similarly, a short paragraph describing the k-d tree in more detail it would be important for readers that are not familiar with the algorithm. Each of the sub-sections 2.1-2.8, could benefit with an expanded explanation on how the algorithm of works.

It is not clear from the manuscript if simple examples for each case (TCs, extra-tropical cyclones, easterly waves) will be available together with the software, so that one can learn to use the software. I would strongly suggest that this would be the case, having full examples, including input files, and examples of output files for the user to reproduce, is fundamental for others to learn to use the software.

---

## Referee Comment (RC2) · Anonymous Referee #2 · 7 Nov 2016

This manuscript is a valuable contribution to storm tracking methodology, with several innovative approaches: tracking on unstructured grids, the use of great-circle distance calculations, and the implementation of a k-d tree algorithm to improve computational efficiency. The article is very well written and researched. I recommend publication with only very minor suggestions.

One of the issues that should be investigated is how well the authors' storm tracking software works on numerical model output with higher horizontal resolution, such as ECMWF 9 km forecasts, which are relied on at many operational weather forecasting centers. In particular, currently deployed storm tracking software often fails to find the center locations of weak, shallow, or highly-sheared tropical cyclones. This presents

a unique opportunity for TempestExtremes. If it proves capable of generating well-behaved, consistent storm tracks for these weak cyclones when other trackers fail, it would be an invaluable tool for the tropical forecasting community.

A question for the authors: Could TempestExtremes be extended to multiple dimensions and used as a coherent feature recognition/object tracking tool? Forecasters sometimes want to track an area of high winds, energy or moisture that rotates around a weather system as the entire system is moving in a general direction. If a contour (closed group of edge points) could be tracked, then this would greatly improve temporally interpolated positions of vortex features, for example.

The authors should be commended for making their tracker software available through GitHub. I agree with other reviewers that a nice step-by-step example with documentation would be a valuable addition to the package and would encourage its use.

---

## Referee Comment (RC3) · Anonymous Referee #2 · 7 Nov 2016

Hi Paul,

You might want to test your TempestExtremes software on weak tropical cyclones in the GFS and other model output then compare it with the forecast aids (tracker output) in the ATCF "a-deck" to quantify its performance.

ftp://ftp.nhc.noaa.gov//atcf/aid_public/

---

## Short Comment (SC1) · 7 Nov 2016

We would like to thank the referee greatly for their positive comments.

To quickly respond to the question for the authors, TempestExtremes does include an experimental capacity to track 2D contours (in our nomenclature we refer to these features as "blobs"). We are currently using this capability to examine atmospheric blocking regions and atmospheric rivers, but agree that it would be advantageous to apply this capability to other features. We are very much open to collaborations that would allow us to further expand the capabilities of the software framework, or see it applied to other problems of interest.

[Figure]

To shed some light on the current capabilities, the StitchBlobs executable takes as input a single NetCDF file (or list of NetCDF files) with an integer variable containing 0s where no feature is present and 1s where a feature is present. It then executes a two step procedure: (1) at each time step uniquely tag distinct blobs of 1s using "flood fill" (also commonly referred to as connected component labelling) and (2) stitch blobs together in time that have any overlap between subsequent timesteps. This procedure must work both forward and backward in time to ensure that, for instance, merging or separating blobs are tagged with the same identifier.

---

## Author Comment (AC1) · 4 Jan 2017

**Reviewer #1.**

This is an excellent manuscript, very well written, in an important topic with many applications. I recommend the publication of the manuscript with minor revisions. My main suggestion is that the authors should expand the explanations of their method, as detailed below. The issue of tracking features in models is an important one, and has important issues. It has many implications, as it hinders model intercomparisons,

as well could have important implications in model projections. Therefore, having an open code that can be used in different models, with different resolutions and for multiple features is extremely attractive. I like the solution of using a great-circle-distance for dealing with models with different resolutions. It would be good for the authors to expand on this topic a little bit, maybe showing an idealized example of with plots of the great-circle distance in models of two different resolutions, to make the solution clearer for the reader. Similarly, a short paragraph describing the k-d tree in more detail it would be important for readers that are not familiar with the algorithm. Each of the sub-sections 2.1-2.8, could benefit with an expanded explanation on how the algorithm of works.

We agree with the reviewer and have added a few additional comments to each of section 2.1-2.8, including a new figure explaining differences between great-circle distance and latitude-longitude distance and a new figure explaining how the $k$-d tree data structure functions.

It is not clear from the manuscript if simple examples for each case (TCs, extra-tropical cyclones, easterly waves) will be available together with the software, so that one can learn to use the software. I would strongly suggest that this would be the case, having full examples, including input files, and examples of output files for the user to reproduce, is fundamental for others to learn to use the software

We also agree with the reviewer on this point and have added three selected examples to the TempestExtremes software package for TCs, ETCs and easterly waves. These can be downloaded from http://climate.ucdavis.edu/tempestextremes_cfsr_tests.tar.gz.

---

## Author Comment (AC2) · 4 Jan 2017

**Reviewer #2.**

This manuscript is a valuable contribution to storm tracking methodology, with several innovative approaches: tracking on unstructured grids, the use of great-circle distance calculations, and the implementation of a k-d tree algorithm to improve computational efficiency. The article is very well written and researched. I recommend publication with only very minor suggestions. One of the issues that should be investigated is

how well the authors' storm tracking software works on numerical model output with higher horizontal resolution, such as ECMWF 9 km forecasts, which are relied on at many operational weather forecasting centers. In particular, currently deployed storm tracking software often fails to find the center locations of weak, shallow, or highly-sheared tropical cyclones. This presents a unique opportunity for TempestExtremes. If it proves capable of generating well behaved, consistent storm tracks for these weak cyclones when other trackers fail, it would be an invaluable tool for the tropical forecasting community.

You might want to test your TempestExtremes software on weak tropical cyclones in the GFS and other model output then compare it with the forecast aids (tracker output) in the ATCF "a-deck" to quantify its performance.

Based on the reviewer's suggestion, we have added a new example entitled "Tropical cyclone forecast trajectories" which uses the algorithm to track 14km CAM output data produced from a simulation of Hurricane Sandy, and compares the results against the NCEP vortex tracker. Overall, the tracker shows consistent performance with NCEP's scheme.

A question for the authors: Could TempestExtremes be extended to multiple dimensions and used as a coherent feature recognition/object tracking tool? Forecasters sometimes want to track an area of high winds, energy or moisture that rotates around a weather system as the entire system is moving in a general direction. If a contour (closed group of edge points) could be tracked, then this would greatly improve temporally interpolated positions of vortex features, for example. The authors should be commended for making their tracker software available through GitHub. I agree with other reviewers that a nice step-by-step example with documentation would be a valuable addition to the package and would encourage its use.

Although TempestExtremes does not have a 3D tracking capability, it does include an experimental capacity to track 2D contours (in our nomenclature we refer to these features as "blobs"). We are currently using this capability to examine atmospheric blocking regions and atmospheric rivers, but agree that it would be advantageous to apply this capability to other features. We are very much open to collaborations that would allow us to further expand the capabilities of the software framework, or see it applied to other problems of interest.

―――――――――――――――――――――